# Piano Performance Evaluation Dataset with Multi-level Perceptual Features

## Abstract

This study aims to build a comprehensive dataset that enables the automatic evaluation of piano performances. In real-world piano performance, especially within the realm of classical piano music, we encounter a vast spectrum of performance variations. The challenge lies in how to effectively evaluate these performances. We must consider three critical aspects: 1) It is essential to gauge how performers perceive and express, and listeners perceive the music, not just the compositional characteristics of music such as beat stability measured from a metronome. 2) Beyond fundamental elements like pitch and duration, we must also embrace higher-level features such as interpretation. 3) Such evaluation should be done by experts to discern the nuanced performances. Regrettably, there exists no dataset that addresses these challenging evaluation tasks. Therefore, we introduce a pioneering dataset PercePiano, annotated by music experts, with more extensive features capable of representing these nuanced aspects effectively. It encapsulates piano performance with a wide range of perceptual features that are recognized by musicians. Our evaluation benchmark includes a novel metric designed to accommodate the inherent subjectivity of perception. For the baseline models, we pinpoint a significant issue in current transformer-based models. We in response introduce a new baseline that leverages hierarchical levels of performance, which shows results comparable to that of large pre-trained models. In conclusion, our research opens new possibilities for comprehensive piano performance evaluation.

## 1 Introduction

In the field of music performance, there exist various performances with the same piece since the same repertoire is constantly reproduced with reinterpretation. It is especially true in the context of classical music, as musicians have been playing e.g. Beethoven for centuries. While this enduring allure adds to classical music's appeal, it also introduces complexity in evaluation since individual performance nuances must be considered in addition to the inherent compositional characteristics of the musical work (Juslin & Laukka, 2004).

It is within this context that human perception has been considered a very important factor in music evaluation. Musicologists have traditionally emphasized perceptive elements in music criticism, which is inevitably subjective. The challenge lies in how to effectively evaluate these features in the various performances. To address this, we must consider two critical aspects of the features and one aspect of musical evaluation. 1) **Perceptual features**: It is essential to gauge how both performers and listeners interpret and express the music. In the context, for instance, evaluating not just the stability of the beat, but also how the pianists accept and convey the rhythm of the music and the way audiences recognize it holds great significance. We delve into the intricate aspects of how the musical features are perceived and expressed in the performance. 2) **Multi-Level features**: Beyond fundamental elements like pitch and duration, we must also embrace higher-level features with a stronger focus on music-oriented and perceptual aspects that are directly connected to human evaluation, such as interpretation. 3) **Evaluation by experts**: Unlike image or language recognition, which often exhibits universality, musicality remains predominantly within the realm of experts (Margulis, 2018). Only experts possess the ability to discern the nuances of music (Czerny, 1879).

However, existing music-related datasets often fall short of addressing these aspects comprehensively, as summarized in Table 1. Many of these datasets neglect the inclusion of perceptual features (Ferraro

|  | source type | various performances | perceptual features | leveled features | expert-annotated |
|---|---|---|---|---|---|
| MAGD | midi, audio | x | x | x | x |
| MAESTRO | midi, audio | o | x | x | x |
| PCD | audio | o | x | x | x |
| GMD | midi | o | x | x | x |
| EMOPIA | midi | x | o | high | o (authors only) |
| Mid-level Perceptual Features | audio | x | o | mid-high | △ (musical test) |
| PercePiano (**ours**) | midi | o | o | low~high | o |

Table 1: Comparison of some existing public music datasets and the PercePiano dataset.

& Lemström, 2018). Even when they do incorporate perceptual aspects, they often do not encompass diverse performances and are inadequate for evaluating performances that inherently exhibit variability (Hung et al., 2021). None of the datasets account for multiple levels of features (Kitahara, 2010; Hung et al., 2021). While some annotations come from experts, their expertise levels can vary due to lower selection criteria determined by a musical test or limited numbers of experts.

To address this challenge, we introduce PercePiano, a novel dataset for expert-guided piano performance evaluation, considering both the perceptual and multi-leveled aspects of music. It encompasses a wide range of musical styles including variations within the same musical piece, thereby capturing the diverse expressiveness of piano performance. Each musical piece is interpreted by 6 to 13 different pianists. Each performance is labeled with comprehensive 19 perceptual features categorized by 9 distinct groups. These labels span aspects ranging from timing to interpretation, each emphasizing the perceptive dimension. Lastly, music experts assessed the performance. Each performance segment received evaluations from 5 to 12 annotators specialized in piano music to gauge consensus in music evaluation. In total, PercePiano comprises 1,202 segments and 12,736 individual notations. Moreover, we have introduced a novel metric into our benchmark, which reflects annotator variance due to the inherent subjectivity in assessing perceptual features. This addition ensures a more reliable and nuanced evaluation process of perceptual elements in piano performances.

We benchmarked several baselines including MusicBERT (Zeng et al., 2021). Notably, we observed that the transformer-based model structure is not suitable to effectively capture the nuanced perceptual features present in piano performances. As part of our study, we introduced a new model for an additional baseline. It explicitly encodes the hierarchy of the performance by anchoring performance data to its corresponding original score data. Our results demonstrate that our new baseline structure outperforms transformer models and performs comparably to pre-trained models. Our work collectively introduces a new concept of piano performance evaluation. Both the complete dataset and baseline code are available after the blind review.

## 2 RELATED WORKS

**Music performance evaluation dataset** In order to evaluate music performance, securing diverse performances including various interpretations and perceptual features of music should be guaranteed. However, not all datasets meet these criteria. MSD Music Genre Dataset (MAGD), for example, which provides genre labels for pop music, lacks both various performances for the same music and perceptual features (Schindler et al., 2012).

Some datasets do not cover perceptual features of each performance. In this case, datasets aim to capture various expressions of performances, including multiple performances of the same musical piece. For instance, the MAESTRO Dataset offers piano performance audio/MIDI data from the Yamaha International Piano-e-Competition [1], ensuring that multiple performances are available for the same piece of music (Hawthorne et al., 2019). Piano Concerto Dataset (PCD) comprises audio source data for piano concertos, featuring recordings from 5 pianists and orchestra (Özer et al., 2023), and Groove MIDI Dataset (GMD) provides expressive drumming performance audio/MIDI data (Gillick et al., 2019).

Even datasets providing perceptual features of music, fall short of encompassing all the essential elements required for a comprehensive evaluation. These datasets primarily focus on higher-level perceptual aspects, neglecting the significance of low-level features, while leveled features were

---

[1] https://www.piano-e-competition.com/

emphasized when analyzing classical piano music (Gerig, 2007; Burton, 2015; Gardner, 2011). For example, the Mid-level Perceptual Musical Features Dataset includes seven mid-level features related to music concepts, whereas the EMOPIA Dataset, which centers on pop piano music, consists of four emotion classes (Aljanaki & Lin, 2022; Hung et al., 2021).

**Leveled features in music**   When it comes to the specific domain of classical piano music, musicologists particularly emphasize features with levels for music analysis (Gerig, 2007). They generally acknowledge that listeners perceive music with fundamental elements such as timbre, through leveled recognition (Gardner, 2011; Burton, 2015). However, in the field of Music Information Retrieval (MIR) or machine learning models related to music, the term "level" holds a distinct meaning. Typically, "low-level features" refer to spectral and cepstral aspects of the music audio signal, while mid-level or high-level representations are consistent with musicology: more music-oriented and perceptive such as genre or emotion (Vatolkin et al., 2014; Aljanaki & Lin, 2022; Kitahara, 2010; Friberg et al., 2014). These classifications do not include low-level perceptual features, such as tempo(timing). From this perspective, there is a demand for datasets that incorporate additional low-level perceptual features with mid-level or higher-level features from existing MIR studies.

**Symbolic music understanding Models**   MIDI-based representation has become a widely used standard for encoding symbolic music (Natsiou & O'Leary, 2021). Recent advancements in MIDI encoding have paralleled those in the field of natural language processing (NLP) owing to its capacity to handle sequential data. Previous works applied Word2Vec to learn embeddings of chords or grouped musical notes from the MIDI corpus (Huang et al., 2016; Hirai & Sawada, 2019). Pre-training approaches (Liang et al., 2020; Zeng et al., 2021; Chou et al., 2021) have emerged afterward, leveraging the power of the transformer's attention mechanisms (Vaswani et al., 2017). Nevertheless, the note-level encoding in transformer-based models falls short for piano performance evaluation because it lacks the consideration of hierarchical structure, which is essential to human perception (Tenney & Polansky, 1980; Terhardt, 1987).

## 3   DATASET

The main goal of our work is to gather diverse performances with leveled perceptual features that match the criteria for performance evaluation. We begin by addressing the challenge of designing labels that effectively encompass the properties (subsection 3.1). In doing so, we meticulously curate leveled features drawing insights from previous research. Additionally, we provide clear guidelines to our annotators to ensure that the features are distinctly "perceptual" in nature. Next, we collect the performances that exhibit such variations and hire only experts to label the data (subsection 3.2).

### 3.1   LABEL DESIGN

To compile the dataset, we employed terminology derived from diverse sources, including traditional musicologists, music information retrieval (MIR), and music AI. We selected a total of 19 features, which were then divided into 9 categories of 4 levels as shown in Table 2. First, we gathered descriptive terms used to portray piano performance. These terms were sourced from piano performance critics (Hinson, 2020; Schonberg, 1987; Park, 1998; Gerig, 2007), writings by pianists themselves (Cooke, 1917; Noyle, 2000), and surveys conducted among Music School students. Next, we selected 19 features from these terms (Option 1, 2 from Table 2) Criteria for feature selection were based on three key considerations: 1) The words are widely used in the music field 2) The selected features avoid extremely positive or negative connotations, maintaining a balanced and neutral perspective 3) The features are not readily extractable from performance data using automated machine methods, indicating their perceptive nature. In other words, what humans can perceive while automated machines cannot.

Then the features were classified into nine categories, based on the musical elements they addressed (Category from Table 2). The selection of these categories was influenced by the works in piano education (Czerny, 1879; 1839), while excluding those that are not directly related to the performance itself (e.g. sight reading, tuning). Lastly, we employed a leveled approach to categorize the musical features, which makes the dataset more explainable. This approach takes into account both the musical aspect (Burton, 2015; Hinson, 2020) and data science or MIR perspective (Kitahara, 2010;

Vatolkin et al., 2014; Aljanaki & Lin, 2022), but primarily following musical taxonomy. In our approach, the levels were divided by the length of music needed to judge each label, as levels for music representation features are often divided into length of analysis window (Kitahara, 2010). Categories that could be detected in the very short term (in the extreme, even with a few notes) are considered as low-level (Timing, Articulation), while high-level category features (Emotion & Mood, Interpretation) should be considered in longer context, at least from musical phrase, sometimes for the whole song. In our criteria, mid-level features are defined as those that could be judged in 2-4 bars length. Features are grouped in level with dotted lines in Table 2.

| Category | Label | Option 1 | Option 2 | D.960 mv2 | D.960 mv3 | D.935 | WoO.80 | ICC(1,1) | ICC(1,k) |
|----------|-------|----------|----------|-----------|-----------|-------|--------|----------|----------|
| Timing | 1 | Stable beat | Unstable beat | 2.96(1.82) | 3.35(1.67) | 3.94(1.71) | 3.31(1.62) | 0.22 | 0.95 |
| Articulation | 2 | Short | Long | 3.33(1.56) | 4.48(1.38) | 3.90(1.49) | 3.63(1.56) | 0.22 | 0.95 |
| Pedal | 3 | Cushioned | Solid | 3.74(1.72) | 4.13(1.46) | 3.68(1.49) | 3.92(1.54) | 0.2 | 0.94 |
|  | 4 | Sparse/dry | Saturated/wet | 2.88(1.59) | 3.99(1.71) | 4.22(1.50) | 3.87(1.46) | 0.42 | 0.98 |
|  | 5 | Clean | Blurred | 4.26(1.84) | 3.59(1.65) | 3.42(1.52) | 3.09(1.37) | 0.33 | 0.97 |
| Timbre | 6 | Even | Colorful | 3.66(1.69) | 3.58(1.44) | 3.84(1.54) | 3.59(1.49) | 0.18 | 0.93 |
|  | 7 | Shallow | Rich | 3.41(1.65) | 4.10(1.44) | 4.03(1.62) | 3.91(1.57) | 0.22 | 0.95 |
|  | 8 | Bright | Dark | 3.82(1.50) | 3.13(1.21) | 3.13(1.31) | 3.99(1.24) | 0.25 | 0.96 |
|  | 9 | Soft | Loud | 3.75(1.61) | 4.48(1.21) | 3.50(1.48) | 3.73(1.52) | 0.22 | 0.95 |
| Dynamic | 10 | Sophisticated | Raw | 3.48(1.60) | 4.02(1.36) | 3.58(1.65) | 3.54(1.59) | 0.24 | 0.95 |
|  | 11 | Little range | Large range | 4.13(1.56) | 4.35(1.33) | 3.83(1.48) | 3.63(1.43) | 0.16 | 0.92 |
| Music Making | 12 | Fast paced | Slow paced | 4.23(1.34) | 3.24(0.99) | 3.40(1.17) | 3.62(1.30) | 0.39 | 0.98 |
|  | 13 | Flat | Spacious | 4.20(1.65) | 3.68(1.30) | 3.90(1.50) | 3.87(1.48) | 0.2 | 0.94 |
|  | 14 | Balanced | Unbalanced | 3.36(1.55) | 3.92(1.30) | 3.98(1.53) | 4.16(1.49) | 0.18 | 0.93 |
|  | 15 | Pure | Dramatic | 3.98(1.74) | 3.90(1.41) | 4.32(1.52) | 4.05(1.58) | 0.17 | 0.93 |
| Emotion | 16 | pleasant | sad | 3.99(1.48) | 3.06(1.21) | 3.13(1.26) | 4.10(1.18) | 0.29 | 0.96 |
|  | 17 | Low Energy | High Energy | 3.96(1.54) | 3.66(1.25) | 4.25(1.21) | 4.02(1.30) | 0.2 | 0.94 |
|  | 18 | Honest | Imaginative | 3.81(1.62) | 4.44(1.20) | 3.84(1.64) | 3.69(1.59) | 0.2 | 0.94 |
| Interpretation | 19 | Unsatisfactory | Convincing | 3.69(1.63) | 4.22(1.37) | 3.64(1.64) | 3.77(1.61) | 0.18 | 0.93 |

Table 2: Statistical Analysis of Piano Performance Evaluation. Each row represents a perceptual feature with 2 opposite options, divided into categories. Musical piece-named columns show mean (*M*) and standard deviation (*SD*) of labeled value (*M(SD)*). ICC(1,1) and ICC(1,k) show intraclass correlation coefficient for single and average ratings.

Such labels have their limitations that, without specific instructions, annotators may tend to focus on technical aspects, particularly low-level features like timing or articulation. To ensure a focus on perceptual features, we designed detailed instructions for annotators, directing their attention solely towards the interpretations made by the pianists, excluding considerations of personal preference or technical aspects. To ensure targeting perceptual features, we designed detailed instructions for annotators to concentrate solely on the interpretations made by the pianists, excluding considerations of personal preference or technical aspects. For instance, we explicitly instructed annotators that "this survey is to examine respondents' perceptions of the performance" and, for each label, such as timing, to consider "how the performer perceives the beat and expresses it in the performance". A comprehensive breakdown of these instructions and annotator requirements can be found in Table 5 and Table 6 in Appendix A.

## 3.2 DATA COLLECTION

To collect performances and labels, we ensure evaluation by experts and diverse performances with different interpretations. We stress that only music expert annotators participated, satisfying one or more of the following conditions: 1) Pianist. Graduate student or higher, or professional performer. 2) Music theory major. College graduates or higher. The music expert annotators were outsourced and evaluated each music segment on a 7-point scale according to each label, assigning 1 point for the strongest agreement with Option 1, 7 points for Option 2, and intermediate values for evaluations in between.

For the choice of the performance, our study focuses on classical piano music. We specifically select pieces from the romantic repertoire, as the romantic era offers expressive, dramatic, and programmatic compositions (Truscott, 1961), providing ample opportunities for performers to shape their interpretations. Two types of musical forms were chosen: piano sonata and variations for solo piano performances. Piano sonata, whose form reached its peak until the romantic era, has to be performed with the individuality of divided movements and a long breath of the whole song (Sandra Mangsen & Griffiths, 2011). Conversely, variations refer to the form of music(or technique) in which the theme is repeated with an altered form of various musical elements. Each variation length is generally shorter than the length of the sonata movements, so individual performances can

show shorter breathing changes. In addition, model learning effectiveness was expected as variations include various techniques of performers (Sisman, 2011)

To gather the performance data, we took advantage of Yamaha's release of performance MIDI files for the entire contest, and the MAESTRO dataset which organized its data (Hawthorne et al., 2019). Considering that Franz Schubert, a prominent composer of the Romantic era, had a round in the competition, we chose Piano Sonata No.21 in B-flat major, D.960 (Schubert, 1828). For variations, 32 Variations in C minor(WoO.80) (Beethoven, 1806), 4 Four Impromptus D.935, No.3 in B-flat major (Schubert, 1827) were selected.

When determining the segment length, previous research used mid-level features of music typically employed segment length of 10 to 30 seconds per song(Vatolkin et al., 2014), and musical phrase, a minimum unit with complete musical sense, typically spans 4 to 8 bars (Nattiez, 1990). For our evaluation purposes, we chose the length of 4 and 8 bars as the segment size, which corresponds to approximately 8 to 30 seconds of playing time. In addition, for comparison, some 16-bar-long segments were also included.

While e-competition and MAESTRO provide high-quality piano performance MIDI data, it does not include the corresponding score of the original piece. To address this, we collected the score files from MuseScore [2], a community-based web platform for musical scores. MusicXML format, XML-based format aiming to represent Western music notation, was used, as it is widely used in MIR research involving musical scores due to its ability to accommodate a wide range of musical characters (Cancino-Chacón et al., 2018; Jeong et al., 2019a; Cuthbert & Ariza, 2010). Indeed, it is important to note that score files obtained from crowdsourced platforms may exhibit inconsistencies in score transcription styles. To ensure accuracy and reliability, we manually reviewed and modified the score files to align with Henle's edition, which is widely regarded as an "Urtext"(undistorted, reliable, and authoritative musical text) edition for classical music pieces (Markevitch, 1997).

### 3.3 DATA ANALYSIS

PercePiano consists of 12,736 annotations for 1,202 musical segments. These segments have lengths of 4 bars (644 annotations), 8 bars (10,283 annotations), and 16 bars (1,809 annotations), and they are derived from performances by 25 human pianists and two types of computer-generated MIDI musical scores. 24 pianists are from e-competition while our recorded pianist is named as numbers, and computer-generated MIDI performances as 'Score' and 'Score2' [3]. 'Score' tends to be a more human-like performance than 'Score2'. The performance of 'Score' includes more musical notations such as dynamics or legato, inducing pianists to play naturally, while 'Score2' has largely removed those elements, making it sound mechanical.

The annotations are ratings scored by 53 different annotators across 19 distinct labels. Specifically, there are 4,076 annotations for Schubert D.960 mv2 (2nd movement), 6,708 annotations for Schubert D.960 mv3 (3rd movement), 644 annotations for Schubert D.935, and 1,308 annotations for Beethoven WoO.80. The average number of annotations per performance segment is 10.52, with a standard deviation of 3.62. Musical piece-named columns (D.950 mv2 to WoO.80) of Table 2 show the mean and standard deviation of the evaluations by the annotators for each performance, based on the 19 features. Detailed statistics for each piece of music and data quality control are introduced in Appendix A.

Inter-annotator agreement is assessed using ICC(Intraclass Correlation Coefficient) (ICC(1,1) and ICC(1,k) of Table 2. Since each segment was annotated by a different set of random annotators, a one-way random evaluation was deemed suitable (Shrout & Fleiss, 1979). The reliability of both single ratings and averages was compared by calculating ICC(1,1) and ICC(1,k) for each label. The results in Table 2 show that ICC(1,1), single measure reliability, is "poor", while ICC(1,k), average reliability, is "excellent" [4]. These findings suggest that individual perspective on music is subjective and may not fully capture the entirety of music, but when averaged, they converge towards a more widely agreed-upon perception of music.

---

[2]https://musescore.com

[3]For Schubert D.960, Score2 is not included.

[4]It is considered to be excellent when the ICC score is over 0.75 and poor under 0.4 (Cicchetti, 1994; Koo & Li, 2016).

# 4 MODELS AND EXPERIMENTS

## 4.1 MODELS

To settle baselines, we view the task as a regression task and aim to predict 19 perceptual features from piano performance MIDI data. As the perceptual features are reliant on human perception, it is crucial to consider the unique aspects of music to enhance the model's ability to evaluate these features. Our contribution to the development of such models can be summarized in two: identifying shortcomings in existing models and introducing a novel architectural approach.

**Challenges for existing models** In the field of music, transformer-based models, like those in NLP, have proven effective in various tasks (Zeng et al., 2021; Chou et al., 2021). Notably, MusicBERT (Zeng et al., 2021) is a powerful pre-trained model for symbolic music understanding. It converts note-level MIDI to OctupleMIDI vector to excel in tasks such as genre and style classification. However, such representation doesn't work well for our objective.

Our first contribution to the baseline is that we point out the problem of the current model regarding PercePiano. Such a transformer structure does not work well when it comes to evaluating perceptual features in piano performance. Our hypothesis is that the MIDI representation that the transformer uses solely relies on note-level information, overlooking explicit consideration of the sophisticated structure in performances.

For music performance, the perceptual boundaries for humans are largely determined by the hierarchically ordered relations of musical elements (Tenney & Polansky, 1980; Terhardt, 1987). Specifically, notes are often organized into voices, which are independent melodic lines or parts within a musical composition. But MusicBERT doesn't leverage the information of higher elements: Notes and voices are grouped into beats, and finally, bars are created by grouping a specific number of beats together. Understanding this hierarchical structure is essential for perceiving music, as it allows us to interpret music beyond individual notes and comprehend the interplay between the different hierarchies.

Though transformer (Vaswani et al., 2017) in NLP captures multiple levels of semantics in the sequential data, such as words, phrases, or sentences (Marcus et al., 1993; Wang et al., 2018), the transformer in the music field fails to capture the semantics. This is primarily due to the inherent noise in timing extracted from performances, whereas the aligned score contains the ground truth. Without properly encoding a hierarchy of musical features, the transformer structure may lead to erroneous interpretations of the hierarchy groups.

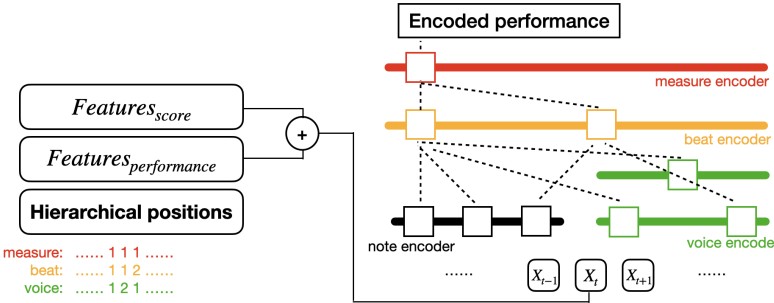

Figure 1: The overview of our model. Each level of performance (measure, beat, voice) is illustrated with different colors (red, orange, green). For simplicity, we only demonstrate the workflow of 3 notes.

**Capturing hierarchical structure** The second contribution is that we design a new baseline, aligning the performance to score data and using hierarchical attention to "ground" the performance for PercePiano. While previous works also explored the concept of alignment in a different context of symbolic music generation (Jeong et al., 2019a;b), annotating alignments has been considered costly. In our target problem of piano performance evaluation, we have the advantage of access to rich and extensive score information. This allows us to effectively align the performance with the score and leverage the hierarchical structure encoded in the score data for improved music understanding.

We introduce our model based on hierarchical attention network (Yang et al., 2016), Percept-HAN (Hierarchical Attention Network for Predicting Perceptual Features), to deal with the challenges involved in perceptual feature evaluation. Figure 1 explains the structure of Percept-HAN. We follow the modeling scheme of the previous works (Jeong et al., 2019a;b) which was originally used to generate symbolic data. The input MIDI data is aligned with the score in MusicXML format to leverage the hierarchical structure. For each note, we calculate its voice, beat, and measure level positions. For the inputs, we use the grounded performance features (e.g. onset deviation, articulation) along with fundamental features from performances (pitch, velocity, start and end time). We also extract the score features such as time signature. All these features are concatenated. If there are misaligned notes or mistouches in the performance, we only extract the performance features and position them according to their nearest notes.

Then, Percept-HAN considers four hierarchical attention levels - note, voice, beat, and bar - to effectively capture the structure of the performance. With an understanding of the hierarchical relations within the music, the model equips a human-like perception of performance. It encodes the aligned data through 4 levels of hierarchy, note, voice, beat, and bar to effectively capture the structure of the score. Each hierarchical unit incorporates a bidirectional LSTM (Bi-LSTM) that encodes each piece of information. The attention network (Vaswani et al., 2017; Yang et al., 2016) processes the output from the lower level to the higher level.

$$u_i = \tanh\left(W_s h_i + b_s\right),$$
$$\alpha_i = \frac{\exp\left(u_i^\top u_s\right)}{\sum_i \exp\left(u_i^\top u_s\right)},$$
$$v = \sum_i \alpha_i h_i,$$

Lower-level embedding is $h_i$. $W_s$ and $b_s$ denote the weight and bias of linear function, $u_s$ denotes lower-level trainable context vector. The output $v$ is the higher-level embedding that encapsulates the information from the lower-level. We set multiple sets of attention heads to enhance the process. Specifically, the output from the note level and voice level is concatenated together to the beat level, then the beat level output is passed to the measure level. We pass the measure level embedding to the feed-forward neural network to get the final predictions per each perceptual feature. We use mean squared error(MSE) as a loss function.

## 4.2 EVALUATION SETTINGS AND MODEL CONFIGURATION

For the evaluation, we randomly leave out roughly 15% of PercePiano as the test set among the 1202 segments. We employ an 8-fold cross-validation strategy to ensure a comprehensive evaluation. We select the best models from each fold regarding the validation set performance and report the average performance on the test set. As multiple users have annotated one MIDI file, we average the annotation values to derive an aggregated value for each feature and normalize them between 0 and 1. We removed labels that are marked as "I don't know" before aggregation.

We utilize two commonly used metrics for the regression task: Mean Squared Error (MSE) and $R^2$. However, it is worth highlighting that there is relatively low agreement among annotators for individual segments, as evidenced by the low ICC(1,1) score presented in Table 2. In light of this, we introduce a novel metric, called range accuracy (RA), that takes into account the standard deviations among annotators. Here's how it works: if the model's prediction falls within the range of $value_{gold} \pm \alpha\sigma_{gold}$, it receives a score of 1; otherwise, it receives a score of 0. The final metric is computed as the average of these scores. In this formulation, $\sigma$ represents the standard deviation, and we experiment with different values of $\alpha$, including 0.1, 0.5, and 1.

We evaluate the model's performance using several baseline models. The first baseline model is **MusicBERT** model (Zeng et al., 2021). In this case, we initialized the model's parameters from pre-trained weights. To specifically assess the structural aspects of the model, independent of pre-training effects, we secondly initialized the MusicBERT model with random weights and referred to it as the **Music Transformer**. For Percept-HAN, we use a public feature extraction tool (Jeong et al., 2019c) [5] to extract aligned input features. All the Bi-LSTMs and the head in each level have

---

[5] https://github.com/jdasam/pyScoreParser

|  | $R^2$ | MSE | RA@1 | RA@0.5 | RA@0.1 |
|---|---|---|---|---|---|
| Music Transformer | 0.43 | 9.18e-3 | 88.71 | 64.49 | 15.41 |
| MusicBERT | 0.55 | 7.30e-3 | 91.36 | 70.78 | 17.80 |
| Percept-HAN | **0.56** | **7.14e-3** | **92.22** | **72.74** | **19.19** |
| w/o score features | 0.54 | 7.51e-03 | 91.76 | 71.55 | 18.95 |
| w/o hierarchical structure | 0.45 | 8.86e-03 | 89.08 | 65.93 | 16.65 |

Table 3: Model results report $R^2$ score and RA at 1/0.5/0.1 (higher the better), and MSE (lower the better) on the test set of PercePiano. We compare the results with pre-trained MusicBERT-small, and Music Transformer which is the MusicBERT initialized with random weights.

a hidden size of 256, which is selected from {64, 256, 512}. The batch size is set to 8, and the learning rate is selected from {5e-5, 2.5e-5}. Also, we implemented the MusicBERT and evaluated its performance on PercePiano on the official repository of MusicBERT [6]. We chose a smaller version of MusicBERT (MusicBERT-small) as it shares a similar parameter count with our model, with both having 17M and 23M parameters, respectively. We set the batch size to 4 and select the learning rate from {1e-5, 5e-6}. We ran a paired t-test using each fold in the cross-validation set to establish a concrete comparison.

## 4.3 MODEL RESULTS

Our experimental results are shown in Table 3. It highlights the effectiveness of our model compared to alternative approaches based on transformer structure. In our first experiment, we assessed the performance of the Music Transformer, which focuses solely on a sequence of individual notes without considering the hierarchical nature of musical performance. Notably, the results consistently yielded $R^2$ scores of 0.43. Percept-HAN significantly outperforms Music Transformer across all metrics ($p < 0.05$). It underscores the limitations of this approach in effectively capturing perceptual features.

Subsequently, we conducted a comparative analysis between our model and MusicBERT. Percept-HAN demonstrates significantly better performance compared with the MusicBERT that relies on pre-training ($p < 0.05$). Detailed results for each feature are provided in Table 7 in Appendix B. Percept-HAN exhibits significant strength across various levels of features. However, MusicBERT underperforms in most features, and their strength is limited to the higher level features such as music making and emotion. This is because capturing low-level features through MIDI data is challenging, as such information is correctly predicted by aligning the performance to the score. Percept-HAN's improved performance stems from its hierarchical comprehension, particularly important for grasping aspects such as timing.

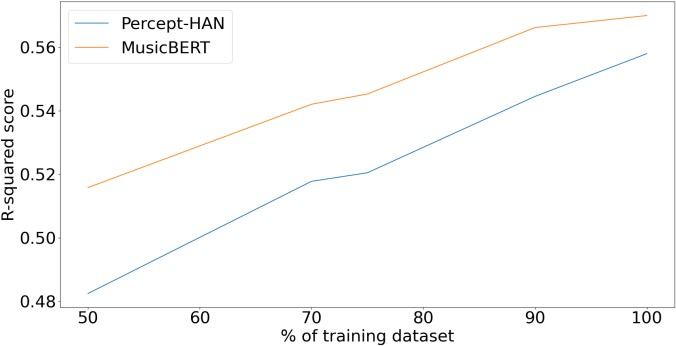

Figure 2: Performance comparison between MusicBERT-base and the Percept-HAN. Pre-training pays off in a low-data regime. However, our approach quickly converges to the performance of MusicBERT as the dataset size grows, showcasing its potential.

In another experiment, we compared our model with the MusicBERT-base. Despite our relatively smaller parameter size (17M vs 103M), Percept-HAN still achieved a superior RA@0.1 score

---

[6]https://github.com/microsoft/muzic/tree/main/musicbert

(18.86 vs 19.19) and showed comparable performance regarding other metrics [7]. Moreover, our experiments revealed that our model shows faster improvement over the growth of dataset size compared to the pre-trained models as shown in Figure 2. While pre-trained models like MusicBERT tend to excel in smaller datasets due to their pre-training, our hierarchical structure-based baseline has shown the potential to catch up and surpass them in bigger datasets. This observation holds particular significance, especially when considering challenges pre-trained models face with domain shifts (Thakur et al., 2021), underlining the advantages of our approach.

With room for optimization still available, we anticipate future research to explore and build upon our findings.

## 5 DISCUSSION

### 5.1 ABLATION STUDY

To understand the contribution of individual components within Percept-HAN, we conducted an ablation study. The results are detailed in the lower part of Table 3. First, we ablated score features and performance features grounded in the score data from the input. The resulting input feature set consists of performance features and position information of each note. This modification led to a slight decrease in performance, from 0.56 to 0.54 regarding $R^2$. Second, we substituted the multi-level hierarchical attention network with note-level Bi-LSTMs. To maintain parameter equivalence, we stacked the Bi-LSTMs from different levels into a single unified model. This alteration had a significantly negative impact across all metrics, emphasizing the model's dependence on explicitly leveraging hierarchical features.

### 5.2 REMAINING ERRORS

Our analysis aimed to identify factors hindering the model's predictive ability by selecting performances with high prediction errors. We selected the performances where the Mean Squared Error (MSE) exceeded 0.1. Through attentive listening and examination, we categorized these performances into distinct scenarios that present challenges for computational models.

First, performances that exhibited disparity in articulation, dynamics, or pedaling degree between the right and left hands proved to be difficult for the model. The model's focus on one hand over the other could lead to differential scoring, adding to the complexity of the learning task. Second, we identified instances where contrasting phrases within a performance complicated the scoring process. If a single piece contained phrases that required distinct articulation or pedal usage, assigning a unifying score presented an inherent difficulty, which the model might have struggled to navigate. Lastly, we observed that overlapping themes within 16-bar pieces, where the performance atmosphere and tempo could abruptly change, posed prediction challenges. To further elevate the precision of our system's performance evaluations, future work should consider expanding the dataset to provide a richer context or refining the selection of musical excerpts for evaluation.

## 6 CONCLUSION

We present PercePiano, a pioneering dataset designed for machine learning-based piano performance evaluation. Annotated exclusively by music experts, PercePiano deals with various levels of perceptual features that are essential in performance evaluation. We suggest a new metric to address the subjectivity issue in evaluating perceptual features. We further study the comprehension of the features and present a new baseline, Percept-HAN. The results highlight the significance of aligning the performance with the hierarchical structure of music for improved comprehension. We anticipate that this line of research opens up exciting possibilities for future systems in the realm of piano performance.

---

[7]The other scores for MusicBERT-base are $R^2$: 0.57, MSE: 6.93e-3, RA@1: 92.74, and RA@0.5: 72.8

## 7 REPRODUCIBILITY STATEMENT

For the dataset, the total number of annotations is in subsection 3.3. We include a detailed description of its statistics and instructions in Appendix A. The evaluation setting with the dataset can be found in subsection 4.2. The model design is described in detail in subsection 4.1. The model configurations with the range of hyperparameters, and the resulting model size are in subsection 4.2. We run paired t-tests to make solid comparisons between models, and the results are in subsection 4.3 and Appendix B.

## 8 ETHICS STATEMENT

In terms of social impact, we expect that the dataset will foster collaboration among musicians, educators, and researchers. Musicians and educators can use the dataset to create educational tools that support learning, while researchers in the field of music performance evaluation can utilize it to develop and validate models.

Our work does not have any ethical concerns. We include a human subject study focused only on annotation tasks conducted through crowd-sourcing. All data contributors have granted informed consent for their work to be included in this dataset. To protect privacy, any personal identifying information has been removed or anonymized. While we have made efforts to ensure the dataset encompasses a broad spectrum of evaluation features and performance variations, it's important to point out certain limitations. There are additional dimensions and nuances within music that are not fully represented in our dataset, and we encourage further exploration in these areas for a more comprehensive understanding.

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

## APPENDIX

## A    QUALITY CONTROL FOR DATASET

This section delineates the data collection process and the quality control procedures implemented for PercePiano.

The entire data collection took place in two rounds of evaluation, with the second round having adjustments based on insights obtained from the initial round. In the first round, we focused on gathering data pertaining to piano sonatas from the Romantic era, specifically targeting the second and third movements of Schubert's D.960. All segments of the entire piece divided into 8-bar units, as well as some 16-bar segments for comparative analysis, were evaluated. Then for the second round, Schubert's D.935 and Beethoven's WoO.80 were selected to represent variations that ensure diverse musical expression.

The most notable difference between the two rounds lies in the number of features used. This disparity arose during the data quality control phase, implemented to enhance dataset reliability given the subjective nature of the measured values. The data filtering process aimed to address two primary concerns: 1) the presence of ambiguous labels that do not exhibit significant differences based on the piece and the performer, and 2) the potential presence of wrongly labeled data.

First, the presence of wrongly ambiguous labels is motivated in Figure 3, which illustrates the mean ratings for Labels 7 (pedal: wet-dry) and 20 (Music making: flexible-inflexible) across the 17 segments of mv2. Each line depicts the evaluation of individual performers. While the assessment of Label 7 clearly exhibits differences between performers, the evaluation of Label 20 does not, which corresponds with the lower $R^2$ values indicated in Table 4. This suggests the existence of labels that are ambiguous for annotators to assess performance.

Therefore, we examined and removed these ambiguous labels from the modeling among the original 28. The labels to be eliminated were identified based on the frequency of ambiguous annotator responses ("I don't know", ex. label 9 > 100) and the lack of significant differences in evaluation values between performances and performers. Specifically, a 3-way analysis of variance (ANOVA) was conducted as the statistical method. The factors in this analysis were movements (mv2, mv3), performers (including 12 pianists, scores), and segments. Due to the differing segment divisions between the 8-bar and 16-bar performances, only the evaluations concerning the 8-bar performances were analyzed. The $R^2$ values in Table 4 serve as indicators of whether the evaluation values significantly differ based on the piece and the performer, with higher values indicating more significance. Labels with smaller $R^2$ values than Label 28 (Interpretation), which presents an overall evaluation of the piece (i.e., $R^2 < 0.23$, adj $R^2 < 0.13$), as well as the sub-items of timing with a high number of missing values (labels 2 - 4) were excluded. Furthermore, although the $R^2$ for Label 27 exceeds the threshold, it was excluded due to its high correlation with Label 26. Therefore, data from only 19 of the original 28 labels were used for modeling. The 19 labels are highlighted in bold in Table 4.

Another effort to ensure dataset quality was to provide precise instructions to participants. As emphasized in the main body, the subjective nature of perceptual features has the risk of lowering the reliability of the dataset by reducing agreement between annotators. We tried to avoid this by providing detailed descriptions for individual categories and features. The contents can be found in Table 5, Table 6.

| Category | Label no. | Option 1 | Option 2 | mv2 $M(SD)$ | mv3 $M(SD)$ | "I don't know" | Adj. $R^2$ | $R^2$ | ICC(1,1) | ICC(1,k) |
|---|---|---|---|---|---|---|---|---|---|---|
| Timing | 1 | **Stable beat** | **Unstable beat** | 2.96 (1.82) | 3.35 (1.67) | 4 | 0.177 | 0.272 | 0.18 | 0.92 |
| | 2 | Mechanical tempo | Natural tempo | 4.87 (1.76) | 4.00 (1.83) | 17 | 0.170 | 0.292 | 0.17 | 0.91 |
| | 3 | Intentional | Unintentional | 3.10 (1.73) | 3.33 (1.62) | 100 | 0.002 | 0.148 | 0.019 | 0.503 |
| | 4 | Regular beat change | Chaotic beat change | 3.46 (1.70) | 3.68 (1.57) | 27 | 0.135 | 0.257 | 0.14 | 0.89 |
| Articulation | **5** | **Long** | **Short** | 3.33 (1.56) | 4.48 (1.38) | 32 | 0.244 | 0.332 | 0.19 | 0.93 |
| | **6** | **Cushioned** | **Solid** | 3.74 (1.72) | 4.13 (1.46) | 27 | 0.157 | 0.254 | 0.15 | 0.91 |
| Pedal | **7** | **Saturated/wet** | **Sparse/dry** | 2.88 (1.59) | 3.99 (1.71) | 54 | 0.476 | 0.537 | 0.44 | 0.98 |
| | **8** | **Clean** | **Blurred** | 4.26 (1.84) | 3.59 (1.65) | 81 | 0.328 | 0.406 | 0.36 | 0.97 |
| | 9 | Subtle change | Obvious change | 3.95 (1.73) | 3.55 (1.57) | 133 | 0.074 | 0.183 | 0.056 | 0.76 |
| Timbre | **10** | **Even** | **Colorful** | 3.66 (1.69) | 3.58 (1.44) | 18 | 0.139 | 0.239 | 0.14 | 0.90 |
| | **11** | **Rich** | **Shallow** | 3.41 (1.65) | 4.10 (1.44) | 12 | 0.203 | 0.295 | 0.18 | 0.92 |
| | **12** | **Bright** | **Dark** | 3.82 (1.50) | 3.13 (1.21) | 31 | 0.250 | 0.337 | 0.25 | 0.95 |
| Dynamic | **13** | **Pure** | **Dramatic/expressive** | 3.98 (1.74) | 3.90 (1.41) | 34 | 0.146 | 0.245 | 0.14 | 0.89 |
| | **14** | **Soft** | **Loud** | 3.75 (1.61) | 4.48 (1.21) | 58 | 0.222 | 0.313 | 0.17 | 0.92 |
| | **15** | **Sophisticated/mellow** | **Raw/crude** | 3.48 (1.60) | 4.02 (1.36) | 22 | 0.176 | 0.272 | 0.17 | 0.92 |
| | 16 | Balanced | unbalanced | 3.36 (1.55) | 3.92 (1.30) | 13 | 0.145 | 0.244 | 0.13 | 0.89 |
| | **17** | **Large dynamic range** | **Little dynamic range** | 4.13 (1.56) | 4.35 (1.33) | 17 | 0.141 | 0.240 | 0.14 | 0.89 |
| Music Making | **18** | **Fast paced** | **Slow paced** | 4.23 (1.34) | 3.24 (0.99) | 17 | 0.412 | 0.480 | 0.39 | 0.97 |
| | 19 | Flowing | Choppy | 3.53 (1.61) | 4.04 (1.38) | 39 | 0.113 | 0.216 | 0.11 | 0.87 |
| | 20 | Swing, flexible | Steady, inflexible | 3.78 (1.54) | 3.35 (1.07) | 60 | 0.120 | 0.222 | 0.11 | 0.87 |
| | **21** | **Flat** | **Spacious** | 4.20 (1.65) | 3.68 (1.30) | 34 | 0.173 | 0.269 | 0.16 | 0.91 |
| | 22 | Harmonious | Disproportioned | 3.36 (1.46) | 3.86 (1.25) | 18 | 0.120 | 0.222 | 0.12 | 0.88 |
| Emotion | **23** | **Optimistic/pleasant** | **Pessimistic/sad** | 3.99 (1.48) | 3.06 (1.21) | 34 | 0.271 | 0.356 | 0.28 | 0.95 |
| | **24** | **High Energy** | **Low Energy** | 3.96 (1.54) | 3.66 (1.25) | 23 | 0.184 | 0.278 | 0.17 | 0.92 |
| | 25 | Dominant, forceful | Subdued | 4.03 (1.41) | 4.06 (1.03) | 67 | 0.091 | 0.197 | 0.079 | 0.82 |
| Mood | **26** | **Imaginative** | **Honest** | 3.81 (1.62) | 4.44 (1.20) | 39 | 0.170 | 0.266 | 0.14 | 0.90 |
| | 27 | Ethereal | Mundane | 4.02 (1.57) | 4.60 (1.15) | 41 | 0.143 | 0.242 | 0.12 | 0.87 |
| Interpretation | **28** | **Convincing** | **Unsatisfactory/doubtful** | 3.69 (1.63) | 4.22 (1.37) | 21 | 0.131 | 0.232 | 0.12 | 0.88 |

Table 4: Statistical Analysis of Piano Performance Evaluation for First-collected Dataset.

| Category | Description |
|---|---|
| All | To define and express the characteristics of individual piano players, this is a survey to investigate respondents' perceptions of the performance. Respondents will listen to the sound sources of various performers' performances for the same song between two eight bars, and respond to the given questions. At this time, it is important to note that this survey is different characteristics of different performers, so you should answer how the performer interprets it compared to a given comparative sample sound source rather than the image of the song itself. For example, rather than responding that romantic songs will be emotionally rich unconditionally, it is necessary to worry and respond to what the performer expresses differently by other performers. As for the questions, please respond by referring to the description of each category. |
| Timing | In this category, questions are asked about the time in the performance, the beat and speed associated with the time. In particular, it contains questions about how the performer perceives the beat and expresses it in the performance. |
| Articulation | It is a category in which the performer asks how to express the articulation. |
| Pedal | We are asking the usage of pedals while performing in this category. |
| Timbre | This category asks questions about the tone itself of the instrument expressed by the performer. It contains questions about the overall change of tone and the degree of abundance. |
| Dynamic | In addition to the overall impression of the tone, this category contains questions that include the degree of intensity, and the flow of time. There is a difference that the intention of the performer is reflected more than the previous category. In other words, it's about the degree to which the player presses the keyboard during the performance and how he/she changes the tone with what intention. |
| Music Making | We ask how the performer made the music in general, and his musicality. Please reflect the wider scope than the previous category, that is, the overall impression of the sound source. |
| Emotion/Mood | This category contains questions about what emotions the performer intended to express in the performance (or how they are delivered to the listener even if they are unintended). Since there are many words that are commonly used to indicate emotional state, this category asks the following two questions. The first is whether the emotional expression of the performance is positive or negative, and you can think of it as bright and dark emotions. The second is whether there is high or low energy revealed in the performance. Depending on these two degrees, we can think of different emotions. For example, when answer was 'positive' and 'high energy', we can say that the emotion of the performance is lively. |
| Interpretation | You may evaluate the performer's interpretation and performance itself. |

Table 5: Description for categories given to annotators.

| Category | Option 1 | Option 2 | Description |
|---|---|---|---|
| Timing | Stable beat | Unstable Beat | Asking whether tempo of the performance is stable, or even. |
| Articulation | Short | Long | Asking whether the performer processes articulation of the performance long/short |
| | Soft, cushioned | Hard, solid | Asking whether aticulation procession is soft and cushioned, or hard and solid |
| Pedal | Sparse/dry | Saturated/wet | Asking of richness in usage of pedals |
| | Clean | Blurred | Question about whether the pedal falls neatly in the performance or it is not easily distinguished and connected |
| Timbre | Even | Colorful | Tone, or sound color of the performance is even in one tone or colorful |
| | Shallow | Rich | Asking richness of the tone |
| | Bright | Dark | Asking brightness of the tone |
| | Soft | Loud | Asking overall dynamics(loudness) of the performance |
| Dynamic | Sophisticated | Raw | Can also called mellow, and crude. Asking how much it is sensitive and subtle for dynamics of the performance |
| | Little range | Large range | Asking of the dynamic range revealed in the performance |
| Music Making | Fast paced | Slow paced | Overall speed of the performance- whether it is fast paced, or slow paced |
| | Flat | Spacious | Asking of overall sense of space that the respondent might feel with the performance |
| | Balanced | Unbalanced | Unbalanced is also called disproportioned. Asking of how it is harmonious, especially for melody |
| | Pure | Dramatic | Dramatic as expressive. Question about the dynamics/energy of the performance |
| Emotion | Pleasant | Sad | Also called Optimistic or Pessimistic emotion. Asking of how much the emotion that felt in the performance is positive(or negative)) |
| | Low Energy | High Energy | Question about the magnitude of emotional energy. For example, even with the same negative emotions, there will be a difference between explosive anger and emptiness. |
| | Honest | Imaginative | Asking the atmosphere felt in terms of imagination. |
| Interpretation | Unsatisfactory | Convincing | Unsatisfactory interpretation also means doubtful. Asking the degree to which the overall interpretation of the performance touches the listener. |

Table 6: Description for each features given to annotators.

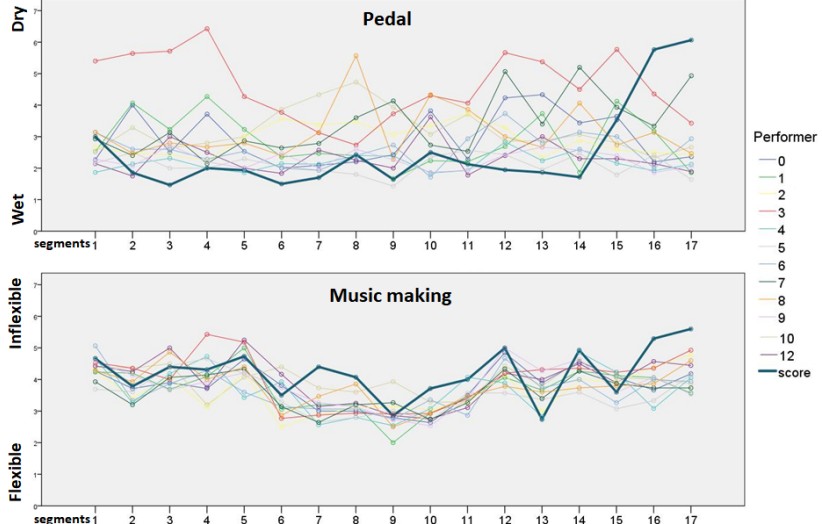

Figure 3: Average evaluation values for the 'Pedal' and 'Music Making' labels for the 17 segments of D960 mv2. Each line represents the evaluation across 13 different performances, with the bold line signifying the evaluation of the musical score performance.

| Category | Label no. | Percept-HAN | | | | | MusicBERT | | | | |
|---|---|---|---|---|---|---|---|---|---|---|---|
| | | RA@1 | RA@0.5 | RA@0.1 | $R^2$ | MSE | RA@1 | RA@0.5 | RA@0.1 | $R^2$ | MSE |
| Total | | **92.22** | **72.74** | **19.19** | **0.56** | **7.14E-03** | 91.36 | 70.78 | 17.8 | 0.55 | 7.30E-03 |
| Timing | 1 | **85.07** | **62.64** | 16 | **0.37** | **1.15E-02** | 81.07 | 57.79 | 13.93 | 0.32 | 1.23E-02 |
| Articulation | 2 | **95.43** | **79.14** | **20.64** | **0.72** | **4.80E-03** | 94.43 | 74.79 | 16.93 | 0.67 | 5.68E-03 |
| | 3 | 94.79 | 78 | **25.29** | 0.62 | 5.68E-03 | **95.79** | 77.36 | 18.14 | 0.63 | 5.61E-03 |
| Pedal | 4 | 84.07 | 58.57 | 14.93 | 0.58 | 1.44E-02 | 84.43 | 57.29 | 13.5 | 0.59 | 1.40E-02 |
| | 5 | 84.64 | 56.43 | 12.86 | 0.38 | 1.57E-02 | 81.14 | 56.64 | 13 | 0.39 | 1.54E-02 |
| Timbre | 6 | 93 | **76.29** | 18.43 | **0.52** | **6.96E-03** | 92.86 | 73.93 | 18.36 | 0.5 | 7.26E-03 |
| | 7 | 91.07 | **73.5** | 20.07 | **0.59** | **7.62E-03** | 91 | 70.5 | 20.36 | 0.56 | 8.29E-03 |
| | 8 | **95.5** | **78.36** | **21.07** | 0.49 | 5.27E-03 | 94.86 | 72.86 | 16.71 | 0.49 | 5.25E-03 |
| | 9 | 96.57 | 79.57 | 22.29 | 0.71 | 4.65E-03 | 96.93 | **81.71** | 20.21 | 0.72 | 4.56E-03 |
| Dynamic | 10 | **93.86** | **73.29** | 18.93 | 0.64 | 5.86E-03 | 90.93 | 71.43 | 19.64 | 0.65 | 5.69E-03 |
| | 11 | 92.71 | 75.43 | 20.5 | **0.46** | **6.40E-03** | 92.36 | 74.07 | 20.14 | 0.4 | 7.05E-03 |
| Music Making | 12 | 91.36 | **69.07** | 14.43 | 0.69 | 4.34E-03 | 91.71 | 65.64 | **16.86** | **0.72** | **3.89E-03** |
| | 13 | 92.21 | **76.14** | 19.07 | 0.55 | 6.31E-03 | 93.5 | 74.21 | 17.07 | **0.57** | **6.07E-03** |
| | 14 | **91.86** | **69.43** | 17.07 | **0.47** | **6.59E-03** | 90.07 | 66.86 | **19.86** | 0.44 | 7.06E-03 |
| | 15 | 91.93 | 74.07 | 21.64 | 0.48 | 7.55E-03 | 91.57 | 73.71 | 22 | 0.47 | 7.68E-03 |
| Emotion | 16 | **94.79** | **72.93** | **20.93** | **0.61** | **4.98E-03** | 91.57 | 67.79 | 15 | 0.52 | 6.14E-03 |
| | 17 | 96.36 | **79.79** | 22.5 | 0.6 | 3.86E-03 | **97.36** | 77.57 | 21.57 | 0.62 | **3.65E-03** |
| | 18 | 93.21 | 75.93 | 20.29 | 0.64 | 5.69E-03 | 92.93 | 75.79 | 19.21 | **0.66** | **5.37E-03** |
| Interpretation | 19 | **93.79** | 73.43 | 17.71 | 0.48 | 7.62E-03 | 91.29 | **74.93** | 15.79 | 0.47 | 7.72E-03 |

Table 7: RA@0.5, RA@0.1, $R^2$ score for each label on the test set of PercePiano. The boldfaced ones are the results that show significant differences ($p < 0.05$).

## B  MODEL PERFORMANCE FOR EACH LABEL

For the detailed comparison, we compared the $R^2$ score between Percept-HAN and MusicBERT-base for individual labels. The results are in Table 7. We boldfaced the results that show significant differences (p < 0.05).

