# OpenReview forum: "PIANO PERFORMANCE EVALUATION DATASET WITH MULTI-LEVEL PERCEPTUAL FEATURES"
_ICLR.cc/2024/Conference — ICLR 2024 Conference Withdrawn Submission_

### Official Review · Reviewer_iYeg · 2023-10-31

**Soundness:** 3 good
**Presentation:** 2 fair
**Contribution:** 2 fair
**Rating:** 3
**Confidence:** 4

**Summary:**

- This paper presents a dataset of piano performances aiming to aid research focusing on he task of piano performance evaluation
- The authors state a few parameters that are important for performance evaluations: 1.) multiple performances of the same piece, 2.) perceptual features which capture both performer’s and listener’s interpretations, 3.) multi-level features which contain low-level details such as pitch/duration of notes and high-level musical concepts, and 4.) expert annotations critiquing the performed music. They note that no existing dataset satisfies these 4 conditions.
- The authors collect a dataset called PercePiano, which includes various performances of the same piece, expert annotations for 19 perceptual features, as well as expert assessments for the performances.
- The authors also present a new model for heirarchical modeling of the performance after aligning it with the score of the piece being performed. The model operates at 4 levels of heirarchy: note, voice, beat, and measure. The proposed model outperforms pre-trained models such as MusicBERT.

**Strengths:**

- The authors have carefully come up with a taxonomy of low-to-high level musical features that encapsulate the quality of a musical performance with annotations collected from experts.
- The presented model seems to be well designed for music performances. Similar ideas have been explored in the past with models like MuseFormer[1], or MeasureVAE[2], but they typically are either constrained to model note-level and measure-level, or have some musical rules baked in, such as attending to specific measures in the past.

[1] Yu, Botao, et al. "Museformer: Transformer with fine-and coarse-grained attention for music generation." Advances in Neural Information Processing Systems 35 (2022): 1376-1388.

[2] Pati, Ashis, Alexander Lerch, and Gaëtan Hadjeres. "Learning to traverse latent spaces for musical score inpainting." arXiv preprint arXiv:1907.01164 (2019).

**Weaknesses:**

- The topic is not too relevant to the audience at ICLR. The hierarchical model may be of some interest but in the current state the paper is not a great fit. A dataset contribution is highly appreciated, however, the ICLR community at large might not benefit too much as compared to a more specialized venue like ISMIR or music perception venues like ICMPC. Conversely, the authors might not gain too much since it will be difficult to foster conversations at a general venue like ICLR where the audience does not have a background in music or music perception.
- The results are not very convincing for the proposed baseline. Since the authors do perform cross-validation, I would be interested in seeing some error-bars for the results. Some of the results look very close and I’m not sure if they are significantly better. I understand that the baseline is not necessarily supposed to be very performant, but given the authors’ justifications for the design it would be expected to perform much better than existing naive transformers.
- Section 4.1 needs a lot more detail. In general I appreciate that the authors spend a lot of time explaining and motivating their design choices and pointing out issues in existing literature. However, this particular section needs specifics about what features are used. The authors simply add a few words in parentheses: (e.g. onset deviation, articulation) and (pitch, velocity, start and end time) for the features used. The paper needs details or at least definitions and a reference about how these are calculated.

**Questions:**

See weaknesses section.

---

### Official Review · Reviewer_vNEe · 2023-11-01

**Soundness:** 3 good
**Presentation:** 2 fair
**Contribution:** 2 fair
**Rating:** 3
**Confidence:** 3

**Summary:**

This paper introduces a novel dataset named PercePiano, designed to provide a comprehensive collection of features and labels for 1202 musical segments. The authors present their motivations for creating this dataset, emphasizing its utility in facilitating a thorough assessment of piano performance. They also provide detailed insights into the data collection process. Moreover, the paper harnesses the Percept-HAN architecture and introduces a set of metrics to assess the effectiveness and practicality of PercePiano.

**Strengths:**

The primary contribution of this paper lies in the introduction of a pivotal dataset, PercePiano, poised to significantly benefit the broader music AI community across various downstream applications, including music transcription, music performance assessment, and music emotion recognition. The authors exhibit a profound foresight regarding the evolving landscape of music AI research, exemplified by their meticulous curation of labels for this dataset. Anticipations are high for the dataset's exceptional quality, promising to set a new standard in the field.

**Weaknesses:**

While the paper makes a commendable contribution in the form of the PercePiano dataset, there are notable concerns regarding its technical novelty and experimental validation. It would be a great contribution if the submission is dataset-target only. However, the proposed metrics, such as MSE, R^2, and RA, lack direct ties to specific downstream tasks, which could limit their effectiveness in assessing the dataset's utility. Furthermore, the model employed in this study has been previously introduced in other works, diminishing its novelty.

It would be better to use several downstream tasks to verify the proposed PercePiano dataset. By establishing several SoTA models and training them with PercePiano can demonstrate its high quality and comprehensive labels and features. For example, there are some tasks can be highly recommended to conduct:

1. The music transcription task. Previously, there exists the GiantMidi dataset with a pianoNet to achieve high performance. Does PercePiano provide more useful labels that can enhance this performance, or extend this transcription task to more dimensions?

2. The music emotion recognition task. Previously, EMOPIA and EMO-Music are main resources for this task. Will the introduction of PercePiano provide better quality?

Additionally, as a dataset paper, it would be great to provide some demos or examples inside PercePiano to fully assess the dataset.

At present, the paper is primarily credited for its dataset contribution, but to make a more substantial impact, it should address these concerns and expand its focus on practical applications and validation.

**Questions:**

The questions are provided in the weakness section above. There is one error in the EMOPIA dataset: it also provides the reference audio tracks.

---

### Official Review · Reviewer_ZEzP · 2023-11-02

**Soundness:** 3 good
**Presentation:** 4 excellent
**Contribution:** 2 fair
**Rating:** 6
**Confidence:** 3

**Summary:**

This paper addresses a gap in the literature regarding performance reception when listening to piano performances. The authors collate a new dataset containing perceptual features collated from expert listeners.

**Strengths:**

* Principled approach to the design of perceptual features to include - this is a well referenced contribution and shouldn’t be overlooked

**Weaknesses:**

* given the dataset is only 3 different songs (though there are a dozen or so performances of each) the claims regarding the model may not be the most robust. Without having to collect more data, perhaps an ablation could be done by cleverly splitting test and train sets of evaluation?

**Questions:**

-

---

### Official Review · Reviewer_Pd1L · 2023-11-04

**Soundness:** 2 fair
**Presentation:** 3 good
**Contribution:** 2 fair
**Rating:** 3
**Confidence:** 4

**Summary:**

This paper presents a newly collected dataset for piano performance evaluation, which focuses on perceptual features. A new model structure is proposed along with a new metric for evaluating perceptual features.

**Strengths:**

This paper presents a new dataset for piano performance evaluation. The strength of the dataset is: 1) it is labelled with a wide range of perceptual features; 2) the annotations are provided by experts.

**Weaknesses:**

1. The dataset exclusively comprises romantic classical piano music, which implies a rather narrow focus and may not be suitable for general piano performance evaluation. This can lead to generalization issue for the models trained on this dataset. In addition, the dataset only contain ~1200 segments, raising questions about the robustness of a model trained on such limited data.

2. In Section 4.1, the authors state that "Such a transformer structure does not work well when it comes to evaluating perceptual features in piano performance". The argument lacks validation. It seems that the problem is associated with the ways of processing input MIDI data rather than the transformer structure. The issue is about the operation of grouping notes and voices into beats and then into bars. And the main novelty of the proposed Percept-HAN is how to leverage hierarchal structure from the input MIDI data. In the proposed structure, Bi-LSTM is used in each hierarchical unit to encodes each piece of information. I'm wondering whether replace the Bi-LSTM with transformer will degrade the performance (if the problem is indeed the transformer structure).

3. The test set is randomly chosen and it's not clear whether the test set contains variations that are not included in the training set. I'm concerned about the generalization ability of the model.

4. All the systems are only evaluated on the PercePiano dataset. It's not clear whether the proposed structure also works with other piano evaluation datasets, especially with a border range of piano music.

5. In Section 4.1, the authors claim that the sophisticated structure is important for evaluating performances. I'm wondering whether 4 bars are enough for the evaluation of such structure. I'm also interested in whether the length of segments affect piano evaluation.

6. Not sure whether ICLR  has the appropriate audience of such dataset and task. Conferences that primarily focus on audio, speech, music may be more suitable for this work.

**Questions:**

1. Table 1 - comparison of dataset should also include the size of dataset (i.e. number of segments, total duration, etc.)

2. The annotation process is not clear to me. In the first paragraph of Section 3.2, it says that the performance is evaluated by experts. In the last paragraph of the section, it says that the scores are collected from MuseScore, a crowdsourced platform, which is confusing.

3. Btw, the opening quotation marks on the third to last line of the first paragraph of section 3.3 are incorrectly written.

---

### Official Review · Reviewer_sviL · 2023-11-05

**Soundness:** 2 fair
**Presentation:** 3 good
**Contribution:** 2 fair
**Rating:** 6
**Confidence:** 3

**Summary:**

Detailed Review to be added in a few hours

**Strengths:**

Detailed Review to be added in a few hours

**Weaknesses:**

Detailed Review to be added in a few hours

**Questions:**

Detailed Review to be added in a few hours

---

### Official Review · Reviewer_zHsN · 2023-11-07

**Soundness:** 2 fair
**Presentation:** 2 fair
**Contribution:** 2 fair
**Rating:** 3
**Confidence:** 4

**Summary:**

This paper presents PercePiano, a new dataset comprised of a sizable body of expert annotations of musical performance quality from the open-source MAESTRO dataset. The authors discuss in depth the dataset curation process. The authors then propose a regression-based performance quality task, wherein a Hierarchical attention-based model is used, beating out transformer-based pretrained baselines on the task.

**Strengths:**

- The overall contribution of the dataset itself is highly valuable. To my knowledge, no widely available datasets exists for music performance assessment that has such fine-grained features, which is of clear interest to the wider MIR community.
- Discussion on dataset creation and cleaning is incredibly thorough and clear.

**Weaknesses:**

I have a few concerns about both the dataset portion and the modeling portion of the work.

**Dataset**
- The authors make the claim that extracted features are chosen with the criteria of "The features are not readily extractable from performance data using automated machine methods, indicating their perceptive nature," but do not go in depth with regards to any evidence of this phenomenon. If there is some more in depth reasoning to support this or related work, it would be important to cite here.
- Constructing all of the features in the binary-sense that is done in the paper feels rather odd and atypical for general description-style features. Namely, it is not clear why the authors did not choose something more established in the literature such as phrasing features as a multi-label tagging problem within each category, as the implied dichotomy between subjective features that the authors used seems arbitrary.
- While resource constraints for annotations are understandable, the fact that the data only comes from a few number of distinct musical pieces limits its usefulness

**Modeling**
- Overall, I have serious concerns about the high level message that transformer-based models *cannot* encode hierarchical information.
    - The authors acknowledge that hierarchical information can be learnt by transformers in the text domain but that "the
transformer in the music field fails to capture the semantics." This comment is made without citation, and seems in direct conflict with the growing progress in the generative music domain (which often uses transformer-based architectures)
    - The claim that MusicBERT does not use hierarchical information seems like a strong overstatement. Namely, just because it does not *explicitly* encode hierarchical information does not mean it doesn't use it *implicility* (the way text-based models do), and would need significantly more experimentation to prove this fact.
    - It is hard to tell whether MusicBERT-small's performance difference described in Table 3 is due to the transformer architecture itself, or rather that Percept-HAN uses "grounded performance features" that MusicBERT never sees in its Octuple encoding. It would be a useful exercise to see whether augmenting MusicBERT with the grounded performance features would perform better or not.
    - Additionally, the note that MusicBERT-base seems to beat the proposed model in all but 1 metric (which should be reported in Table 3) seems to directly contradict the claim that "transformers do not encode hierarchical information," and thus the messaging could be changed to reflect that any claims of transformers lack of hierarchical reasoning are constrained to small models.
- I am concerned there may be severe data leakage issues in the entire experimental set-up. Namely, it is not clear whether training and validation data are split according to song or not. If not, it is hard to trust any of the evaluation metrics, as there may be considerably high correlation between features from one part of a piece to another. If possible, the authors should rerun experiments making sure to minimize the conceptual overlap between training and validation sets (such as restricting one song to only occur in the validation set).

**Questions:**

- The explanation of Percept-HAN is a bit hard to follow. Could you describe more in depth specifically how the model encodes features from multiple hierarchical levels?